# Enhancement of Peroxydisulfate Activation for Complete Degradation of Refractory Tetracycline by 3D Self-Supported MoS_2_/MXene Nanocomplex

**DOI:** 10.3390/nano14090786

**Published:** 2024-04-30

**Authors:** Yuxia Song, Runhua Chen, Shihai Li, Shali Yu, Xiaoli Ni, Minglong Fang, Hanyun Xie

**Affiliations:** College of Life and Environmental Sciences, Central South University of Forestry and Technology, Changsha 410004, China; yuxiasong@csuft.edu.cn (Y.S.);

**Keywords:** MoS_2_, MXene, PDS, tetracycline, ROS

## Abstract

Antibiotic abuse, particularly the excessive use of tetracycline (TC), a drug with significant environmental risk, has gravely harmed natural water bodies and even posed danger to human health. In this study, a three-dimensional self-supported MoS_2_/MXene nanohybrid with an expanded layer spacing was synthesized via a facile one-step hydrothermal method and used to activate peroxydisulfate (PDS) for the complete degradation of TC. The results showed that a stronger •OH signal was detected in the aqueous solution containing MoS_2_/MXene, demonstrating a superior PDS activation effect compared to MoS_2_ or Ti_3_C_2_T_X_ MXene alone. Under the conditions of a catalyst dosage of 0.4 g/L, a PDS concentration of 0.4 mM, and pH = 5.0, the MoS_2_/MXene/PDS system was able to fully eliminate TC within one hour, which was probably due to the presence of several reactive oxygen species (ROS) (•OH, SO_4_^•−^, and O_2_^•−^) in the system. The high TC degradation efficiency could be maintained under the influence of various interfering ions and after five cycles, indicating that MoS_2_/MXene has good anti-interference and reusability performance. Furthermore, the possible degradation pathways were proposed by combining liquid chromatography–mass spectrometry (LC-MS) data and other findings, and the mechanism of the MoS_2_/MXene/PDS system on the degradation process of TC was elucidated by deducing the possible mechanism of ROS generation in the reaction process. All of these findings suggest that the MoS_2_/MXene composite catalyst has strong antibiotic removal capabilities with a wide range of application prospects.

## 1. Introduction

The content of refractory and toxic antibiotics in water bodies is increasing due to the random dumping of personal care products (PPCPs), active pharmaceutical compounds, and other medical supplies in recent years, posing a serious threat to the ecological environment of natural water bodies and even human health [1,2]. Among them, tetracyclines (TCs) are widely used in medicine and animal husbandry due to their low cost and broad-spectrum activity, resulting in significant environmental risks [3,4]. It is difficult to remove TC from the environment using common methods. For instance, adsorption can merely transfer TC in water to the surface of a solid adsorbent without removing it fundamentally, while biological resistance and chemical stability would limit the biodegradation process, resulting in a low degradation efficiency (below 56%) [5,6]. Therefore, an innovative method for eliminating TC from the environment is critical.

Applying advanced oxidation processes (AOPs) to the oxidative destruction of organic compounds is a major area of current research. When compared to the traditional hydroxyl radical (•OH), the sulfate radicals (SO_4_^•−^) of sulfate radical-based advanced oxidation processes (SR-AOPs) have a greater redox potential (2.5–3.1 eV vs. 1.8–2.7 eV), a longer half-life (30–40 s vs. 20 ns), a wider pH range, and higher selectivity, providing them an edge in degrading organic contaminants [7,8]. Nevertheless, only when activated under certain conditions (such as UV light [9], ultrasound [10], carbon-based materials [11], and transition metals [12]) that peroxydisulfate (PDS) can produce reactive oxygen species (ROS), including sulfate radicals (SO_4_^•−^) and superoxide radicals (O_2_^•−^), to properly breakdown organic contaminants [13]. However, UV radiation and ultrasonic waves are limited by their high cost and low energy efficiency, while carbon-based catalysts can only adsorb organics to their surface rather than decompose them [14,15]. It has been reported that transition metals could be useful catalysts for PDS activation [7]. Generally, Co^2+^ and Fe^2+^ are the most widely utilized transition metals, and they have a high ability to activate PDS [16]. Unfortunately, limitations such as high cost and toxicity from Co^2+^ leaching and the conversion of Fe^2+^ to Fe^3+^ to produce precipitation would interfere with PDS activation, restricting the application of these two ions in practice [16,17].

Layered transition metal dichalcogenides (TMDs, generally MoS_2_) have been reported to be employed as inorganic activators or catalysts to activate PDS (S_2_O_8_^2−^) to generate ROS directly for pollutant disintegration [16,18]. Furthermore, MoS_2_ is an exceptional chemical reagent for effectively activating PDS, owing to its low cost, excellent chemical stability, mechanical robustness, non-toxicity/low toxicity, and flexible physical/chemical properties [19]. However, the native interlayer spacing of MoS_2_ is rather small (0.3 nm), limiting the capacity of ions/molecules to interact with internal S atoms and impeding the efficient activation of PDS by active surfaces and edge sites [17,20]. Therefore, MoS_2_ nanocomposites with an enlarged interlayer spacing are required. It has been proven that solvothermal synthesis with Mo and S precursors can produce MoS_2_ nanosheets with an increased interlayer spacing (0.94 nm), along with a surface-exposed plentiful active group, allowing them to exhibit stronger activity [17]. Additionally, MoS_2_ nanosheets are straightforward to aggregate since their intrinsic high surface energy results in a drop in electron transfer rate and surface-active sites [21]. As a result, directional growth of MoS_2_ on a hydrophilic, large, specific surface carrier is another effective strategy for increasing layer spacing, thereby exposing more active sites and improving its application in wastewater treatment [22]. In recent years, novel MXene materials (such as Ti_3_C_2_T_X_ MXene) have attracted extensive attention in various fields due to their excellent performance [23,24]. MXene, a two-dimensional transition metal substance consisting of selectively etched ternary nitrides or carbides, with ecologically favorable features, a large surface area, excellent chemical stability, unique metallic thermal/electrical conductivity, and hydrophilic qualities, has been widely used in sensors, lithium-ion batteries, supercapacitors, and environmental applications [25]. It is mainly composed of transition metal carbides, and the chemical formula is commonly expressed as M_n+1_X_n_T_x_, where M is a transition element (e.g., V, Mo, Ti, etc.), X is C or N, and T_x_ stands for the surface terminations. Among the various MXene materials, Ti_3_C_2_T_x_ is the most widely used [24]. However, the fabrication of devices and functional coatings based on Ti_3_C_2_T_X_ remains challenging as they are susceptible to chemical degradation through oxidation to TiO_2_ [24,26,27]. In addition, the hydrothermal/solvothermal techniques that are typically used to produce MXene-based nanomaterials can accelerate MXene oxidation. It is found that Ti_3_C_2_T_X_ MXene is easily oxidized during the hydrothermal process due to the presence of dissolved oxygen, resulting in performance loss [28]. Consequently, a simple solution approach is urgently needed to completely encapsulate and anchor MoS_2_ nanosheets on the surface of Ti_3_C_2_T_X_ MXene to combine the useful properties of these two components into functional heterojunctions and overcome their respective deficiencies for efficient activation of PDS for catalytic degradation of TC. 

In this study, we designed an effective and stable MoS_2_/MXene catalyst for the successful activation of PDS to generate ROS to catalyze the breakdown of residual refractory TC in water. Furthermore, MoS_2_ nanoflowers were strongly coupled with Ti_3_C_2_T_X_ MXene via a well-designed one-step hydrothermal process. In this method, a new sulfur source, sodium diethyldithiocarbamate trihydrate (DDC), and a capping agent, EDTA, were used to substantially widen the interlayer spacing of the nanocomposite and to gain abundant surface-active sites, while the excellent chelating ability of DDC allowed the MoS_2_ precursor to be tightly coupled to Ti_3_C_2_T_X_ MXene, thus effectively preventing the chemical reaction of Ti_3_C_2_T_X_ MXene at the initial stage of the hydrothermal reaction [23,29]. The synthesized composites were characterized and analyzed using SEM, FTIR, and XPS, and the effects of different factors on the degradation of TC in water by MoS_2_/MXene-activated PDS were meticulously investigated, while also inferring the possible degradation mechanism of TC. The findings of this study will contribute to the construction of MoS_2_, MXene, and their composites, as well as the production of powerful persulfate-activating catalysts, allowing for the efficient purification of refractory persistent organic pollutants in water.

## 2. Materials and Methods

### 2.1. Materials

Titanium aluminum carbide (Ti_3_AlC_2_) was obtained from 11 Technology Co., Ltd. in Jilin, China. Sodium molybdate dihydrate (Na_2_MoO_4_·2H_2_O), sodium diethyldithiocarbamate trihydrate (DDC), ethylenediaminetetraacetic acid (EDTA), tetracycline hydrochloride (TC), sodium persulfate (Na_2_S_2_O_8_, peroxydisulfate (PDS)), and p-benzoquinone (BQ) were purchased from MACKLIN Biochemical Co., Ltd., Shanghai, China. Sulfuric acid (H_2_SO_4_), sodium hydroxide (NaOH), lithium fluoride (LiF), anhydrous ethanol (C_2_H_5_OH), sodium chloride (NaCl), sodium nitrate (NaNO_3_), sodium dihydrogen phosphate (NaH_2_PO_4_), sodium carbonate (Na_2_CO_3_), and tert-butyl alcohol (TBA) were purchased from Sinopharm Chemical Reagent Co., Ltd., Shanghai, China. All chemicals used are of analytical grade, and deionized water was used in all experiments.

### 2.2. Synthesis of MoS_2_/MXene

The facile one-step hydrothermal synthesis method of MoS_2_/MXene is shown in Figure 1. Firstly, the preparation of Ti_3_C_2_T*_x_* MXene dispersion followed the method described in previous studies [28,30]. A standard hydrothermal procedure was used to synthesize MoS_2_/MXene nanocomposites. Briefly, 0.7566 g of sodium molybdate dihydrate and 0.2 g of EDTA were dispersed in 50 mL of the pre-prepared Ti_3_C_2_T*_x_* MXene dispersion, which underwent absolute ultrasonic processing for 1 h. After oscillation mixing for 0.5 h, 0.7650 g of DDC was added to the suspension with continuous mixing for 0.5 h. Subsequently, the combination was heated at 180 °C for 24 h in a Teflon-lined hydrothermal autoclave reactor. Upon cooling the solution to room temperature naturally, the resulting precipitate was washed several times with deionized water/absolute ethanol, and then, the MoS_2_/MXene complex was collected after freeze-drying under vacuum. MXene and MoS_2_ monomers were prepared under the same conditions for subsequent analysis and comparison as the blank control. In addition, several MoS_2_/MXene samples were prepared by setting different mass ratios of MoS_2_ and MXene (1:1, 2:1, and 3:1).

### 2.3. Characterization

A variety of characterization approaches were used to observe the microscopic surface arrangement and element valence makeup of the prepared MoS_2_/MXene samples. The morphology and structure of different samples were observed using a scanning electron microscope (SEM; ZEISS Sigma 300, Roedermark Germany) equipped with an energy-dispersive X-ray spectrometer (EDX). The chemical bonds and surface functional groups of MoS_2_, MXene, and MoS_2_/MXene were investigated via Fourier-transform infrared spectroscopy (FTIR; Thermo Scientific Nicolet iS20, MA, USA). The surface elemental composition and chemical state were analyzed using X-ray photoelectron spectroscopy (XPS; Thermo Scientific K-Alpha, MA, USA).

### 2.4. ESR Analysis

To better illustrate the role of reactive oxygen species in the activation of PDS by MoS_2_/MXene, electron spin resonance (ESR) spectra were obtained using an ESR spectrometer (JESFA200, JEOL) in air, with a resonance frequency of 9225.960 MHz, a microwave power of 0.998 mW, a center field of 329.4 mT, a modulation frequency of 100 kHz, a sweep width of 5 × 1 mT, a time constant of 0.03 s, and a sweep time of 30 s, and 5,5-dimethyl-1-pyrroline -N-oxide (DMPO) was used as a spin trap to capture hydroxyl radicals and generate DMPO-•OH signals using hyperfine splitting at room temperature.

### 2.5. TC Degradation Experiments

All TC degradation experiments were carried out in 150 mL Erlenmeyer flasks at 25 °C under varying experimental conditions, which included the material system (MXene only, MoS_2_ only, and MoS_2_/Mxene), catalyst dosage, PDS consumption, pH, and initial concentration of TC.

Specifically, 50 mL of TC solution was placed in a 150 mL flask. After a certain amount of the catalyst material was added to the solution, which was stirred for 10 min, PDS of a certain concentration was added to initiate the degradation reaction. During the reaction process, 1.0 mL of the reaction solution was withdrawn at specific time intervals (10, 20, 30, 40, 50, and 60 min) with filtering. The recovered solution was filtered through a 0.22 μm filter and diluted to the standard curve concentration range with deionized water. The residual concentration was then detected at 356 nm using a UV–visible spectrophotometer (UV-2700; Shimadzu, Kyoto, Japan).

Additionally, (1) the impacts of inorganic anions (Cl^−^, NO_3_^−^, H_2_PO_4_^−^, and CO_3_^2−^), which are frequently present in natural water, were investigated in this work at a particular concentration of TC solution, a certain mass of catalyst, and a given molar mass of PDS solution. (2) Different concentrations of EtOH, TBA, and BQ were used as quenchers to study their effects on the oxidative degradation of TC.

## 3. Results and Discussion

### 3.1. Synthesis and Characterization of MoS_2_/MXene

According to our previous reports, MXene is prone to oxidation at high temperatures and has poor stability features, which limits its use to some extent [28]. In this study, MXene, MoS_2_, and MoS_2_/MXene nanomaterials were prepared under the same conditions, and it can be seen from the SEM image in Figure 2a that the MXene surface was oxidized at a high temperature, and a significant amount of TiO_2_ grew on the surface of the initially smooth MXene sheet [21]. The pure MoS_2_ nanosheets were connected in clusters and dispersed in a petal-like pattern in all directions, as shown in Figure 2b,c. The SEM images of MoS_2_/MXene at different magnifications are presented in Figure 2d–f. It was observed that numerous MoS_2_ nanoflowers were uniformly growing on the MXene surface. With further magnification, it could be clearly observed that an intercalation structure between MoS_2_ and MXene was formed, and the disappearance of TiO_2_ particles on the MXene surface suggested that the addition of MoS_2_ partially shielded the oxidation of MXene. The EDX mapping image of MoS_2_/MXene is displayed in Figure 2g, and the results showed that the elements C, O, Mo, S, and Ti were uniformly distributed. The results of the elemental composition analysis via EDX spectroscopy are shown in Appendix A. It was determined that the atomic ratio of Mo to S was about 1:2, suggesting effective synthesis of MoS_2_, and the high contents of C and O indicated the possibility of the presence of abundant organic functional groups. The SEM results indicated that MoS_2_/MXene composite materials with a stable 3D intercalation structure were successfully prepared.

As shown in Figure 3a, after being exposed to high temperatures, MXene exhibited a broad absorption peak at 3300–3500 cm^−1^ and a significant absorption peak at 450–530 cm^−1^, which were attributed to the stretching vibration of the O-H single bond and the Ti-O-Ti bond, respectively [31]. It indicated that the MXene surface was readily oxidized to generate TiO_2_ under high temperatures. Additionally, the disappearance of the characteristic peak of the Ti-O-Ti bond was observed in the FTIR chart of MoS_2_/MXene, indicating that MoS_2_ could serve as a reactive oxygen barrier for MXene and shield it from oxidation at high temperatures [32]. The characteristic peak at 2926 cm^−1^ was due to -CH_3_ stretching [33]. The characteristic peaks at 1055, 1123, 1541, and 1629 cm^−1^ were assigned to the stretching vibrations of the C-N, C-O, N-H, and C=O bonds of the EDTA fraction attached to the MoS_2_ surface, respectively [34]. Furthermore, the peaks at 765 and 596 cm^−1^ corresponded to out-of-plane C-H bond bending and S-S bond stretching, respectively, due to bond formation between the S atoms of MoS_2_ and DDC [35]. The characteristic peak at 457 cm^−1^ was associated with S-Mo stretching, indicating the successful formation of MoS_2_ nanostructures [29,36]. Apparently, MoS_2_/MXene possesses more functional groups and surface-active sites, which are potentially more conducive to the effective activation of PDS for the degradation of various refractory organics.

To further investigate the chemical state and composition of the synthesized MoS_2_/MXene nanocomplexes and to identify the functional group species, an XPS examination was performed. As shown in the measured spectra (Figure 3b), the peaks at 532.2, 456.1, 285.1, 228.4, and 161.8 eV corresponded to O 1s, Ti 2p, C 1s, Mo 3d, and S 2p, respectively [29,37]. The results showed that MoS_2_ and MXene were successfully combined, which is consistent with the findings of the SEM and FTIR analyses. In addition, in the high-resolution spectrum of C1s (Figure 3c), the peaks at 281.4 eV, 284.8 eV, 286.4 eV, and 288.8 eV corresponded to Ti-C, C-C, C-O, and O-C=O, respectively [38]. In the O1s high-resolution spectrum (Figure 3d), the peaks at 529.7 eV, 532.0 eV, and 533.6 eV were associated with Ti-O, C-OH, and C=O, respectively [39]. All of these characterization findings suggested that there were several functional groups on the composite surface. In the Mo 3d spectrum (Figure 3e), the double peaks had lower binding energies (231.5 and 228.3 eV), which were associated with Mo^4+^ 3d_3/2_ and 3d_5/2_ of MoS_2_, respectively, and the weak characteristic peaks at 233.1 and 235.6 eV were caused by the presence of trace quantities of unreduced Mo^6+^ [29,40]. Additionally, the peak detected at 225.3 eV could be attributed to the S 2s signal [20]. In the S 2p spectrum (Figure 3f), the obtained double peaks of S 2p_3/2_ and S 2p_1/2_ at ~161.0 and ~162.2 eV were attributed to the S^2−^ of MoS_2_ [29]. These analytical test findings showed that MoS_2_ and MoS_2_/MXene with plenty of edge active sites were successfully prepared.

### 3.2. Performance Regarding TC Degradation

The effects of different materials and systems, catalyst dosage, PDS consumption, pH, and initial concentration of TC on its degradation were investigated, and the results are shown in Figure 4. 

Since the best TC removal effect was obtained when the mass ratio of MoS_2_ to MXene was 3:1, the composites with this mass ratio were used in subsequent experiments (Appendix A). As shown in Appendix A, when MXene or MoS_2_ was present alone, the degradation efficiency of TC was limited (41.28% and 55.56%, respectively), while the degradation efficiency of TC became higher (75.30%) when MoS_2_/MXene was used.

As shown in Figure 4a, the effects of different materials or systems on TC degradation were studied under the same catalyst dosage or PDS concentration conditions. In the presence of PDS alone for 1 h, the degradation efficiency of TC was poor (18.07%), while the addition of MXene or MoS_2_ accelerated the degradation efficiency of TC (71.82% and 74.06%, respectively). More interestingly, the inclusion of MoS_2_/MXene significantly accelerated the degradation of TC, which could achieve 100% efficiency at around 50 min. In addition, it can be inferred from Appendix A that all four systems conformed to the pseudo-first-order kinetic equations when degrading TC, and the parameters are listed in Appendix A. These results imply that the vertical development of MoS_2_ on MXene exposes more active sites, thus enhancing electron mobility across the composites and promoting the rapid degradation of TC [41].

The impacts of different dosages of catalyst and PDS concentrations on TC degradation were also investigated, and the results are displayed in Figure 2b,c. The TC degradation efficiency improved significantly from 57.45% to 100% within 1 h of the process as the catalyst dosage rose from 0.05 g/L to 0.4 g/L. This was attributed to an increase in the rate of production of •OH and other reactive oxygen species (ROS) inside the reaction system as the MoS_2_/MXene concentration increased. However, despite the increase in catalyst dosage, the degradation efficiency of TC did not improve considerably. This could probably be attributed to the limited number of active sites on the catalyst [42]. Additionally, at a catalyst dosage of 0.4 g/L, the impact of varying PDS concentrations on TC removal was investigated. The rate of TC degradation exhibited a general trend toward improvement as the PDS concentration increased (75.88%→100%). It might be attributed to an increase in the concentration of PDS in the MoS_2_/MXene/PDS system, which promoted the excitation of S_2_O_8_^2−^ by MoS_2_/MXene to form SO_4_^•−^ while also producing more active species, thereby efficiently catalyzing the breakdown of TC in the solution [43].

Similarly, the effectiveness of TC degradation was also influenced by the initial pH of the reaction system. As shown in Figure 4d, the degradation efficiency of TC under weakly acidic and neutral circumstances was greater than that under acidic and weakly basic conditions. When the initial pH was raised from 3.0 to 5.0, the degradation efficiency of TC rose from 90.56% to 100%, but as the pH was increased further from 7 to 9, the degradation efficiency was inhibited at a rate of 1.09% and 10.36%, respectively. This may be explained by the fact that TC in an aqueous solution occurs as cations under an acidic environment, whereas the MoS_2_/MXene surface is readily protonated and positively charged, which causes their mutual exclusion and reduces the effectiveness of removing TC [44]. Moreover, the difference in TC degradation efficiency may be attributed to the fact that when the initial pH is too low, a considerable quantity of H^+^ in the solution will form hydrogen bonds with the O-O group in S_2_O_8_^2−^, which negatively impacts PDS activation or is associated with the autolytic degradation of HSO_5_^•−^ and the generation of SO_5_^•−^ [45,46]. On the other hand, TC is more easily adsorbed on the surface of the catalyst in weakly acidic or neutral conditions, leading to a higher PDS activation efficiency, and thus, improving the degradation efficiency [47]. In contrast, because TC in an aqueous solution exists as anions under alkaline circumstances and the material’s surface protonation is impaired, its adsorption is diminished, and a considerable amount of OH^−^ in the solution competes with TC molecules, thereby limiting their removal [48]. In addition, the impact of the initial concentration of TC was investigated. The results clearly showed that when the initial concentration of TC increased, its degradation efficiency dropped since increasing the initial concentration in the reaction system meant more reactive oxygen species (ROS) were present for catalytic degradation [49].

To further explore the action pattern of •OH production by MoS_2_/MXene nanoparticles during TC degradation, ESR was employed to detect the DMPO-•OH signal intensity, and the results are shown in Figure 5. It was demonstrated that MoS_2_/MXene nanoparticles generated significantly more •OH in acidic air than in neutral or basic air (Figure 5a–c). Due to the intermolecular polarization of MoS_2_, the MoS_2_/MXene nanocomposite created a piezoelectric potential to stimulate the breakdown of water molecules to form •OH, which was accelerated at higher H^+^ concentrations [50]. The results clearly demonstrated that MoS_2_/MXene could produce •OH autonomously in exposed aqueous solutions. Additionally, the intensity peak of DMPO-•OH dropped with an increase in solution pH, showing the strongest signal peak at pH = 4.0 compared to other solution pH values (7.0 and 9.0). The intensity of the DMPO-•OH signal is shown in Figure 5d,e, when MXene and MoS_2_ were employed as an independent catalyst, respectively. The results revealed that under the same circumstances, the intensity of the DMPO-•OH signal was significantly lower than that of the MoS_2_/MXene nanocomposite when only MXene or MoS_2_ was present. Therefore, it could be inferred that the formation of particular heterojunctions between MoS_2_ and MXene enhanced the nanocomposite’s ability to transmit electrons, thus promoting the generation of •OH [41,51], which potentially stimulated PDS activation and enhanced TC degradation in turn.

### 3.3. Effect of Inorganic Anions

The impacts of inorganic anions (Cl^−^, NO_3_^−^, H_2_PO_4_^−^, and CO_3_^2−^), which are frequently present in natural water, were investigated in this work. As shown in Figure 6a, a relatively low concentration (0.5 mM) of Cl^−^ could significantly reduce the removal efficiency of TC. Moreover, increasing the concentration of Cl^−^ from 0.5 mM to 1.0 mM enhanced the degradation efficiency of TC, although a further increase would restrict the degradation efficiency. This is probably due to the fact that Cl^−^ at low concentrations would combine rapidly with SO_4_^•−^ to form SO_4_^2−^, which would hinder the degradation of TC in an aqueous solution, whereas higher concentrations might generate other active species such as Cl^•^ and Cl_2_^•^, which might stimulate TC breakdown (Equations (1) and (2)) [52]. However, when the concentration of Cl^−^ was gradually increased to 20 mM, various active species were depleted, and some inorganic compounds were generated in the system (Equation (3)) [53]. Therefore, the degrading efficiency of TC was slightly impacted.

Similarly, when the concentration of NO_3_^−^ was raised from 0.5 mM to 10 mM, the degradation of TC reduced from 26.55% to 14.66% (Figure 6b), which could be attributed to the generation of NO_3_^•^ (Equation (4)) [54]. However, when the concentration of NO_3_^−^ was increased further, the degradation rate of TC reduced, which might be related to the conversion of various highly active chemicals into less active or inactive ones in the system.

In contrast, the degradation rate of TC decreased with an increase in the concentrations of H_2_PO_4_^−^ and CO_3_^2−^ (Figure 6c,d), which might be due to the scavenging action of these inorganic anions on SO_4_^•−^ (Equations (5)–(10)) [55]. Overall, as shown in Figure 6e, the order of the inhibition effect of inorganic anions was CO_3_^2−^ > H_2_PO_4_^−^ > NO_3_^−^ > Cl^−^, which was essentially consistent with the conclusions of previous studies [56].
(1)$${\mathrm{Cl}}^{-}+SO_{4}^{2-}\to SO_{4}^{2-}+{\mathrm{Cl}}^{•}$$
(2)$${\mathrm{Cl}}^{-}+{\mathrm{Cl}}^{•}\to {\mathrm{Cl}}_{2}^{•}$$
(3)$${2\mathrm{Cl}}^{-}+S_{2}O_{8}^{2-}\to 2SO_{4}^{2-}+{\mathrm{Cl}}_{2}$$
(4)$${{\mathrm{NO}}_{3}}^{-}+SO_{4}^{•-}\to SO_{4}^{2-}+{\mathrm{NO}}_{3}^{•-}$$
(5)$${{\mathrm{CO}}_{3}}^{2-}+SO_{4}^{•-}\to SO_{4}^{2-}+{\mathrm{CO}}_{3}^{•-}$$
(6)$${{\mathrm{CO}}_{3}}^{2-}+H_{2}O\to {\mathrm{OH}}^{-}+{{\mathrm{HCO}}_{3}}^{-}$$
(7)$${{\mathrm{HCO}}_{3}}^{-}+SO_{4}^{•-}\to SO_{4}^{2-}+{{\mathrm{HCO}}_{3}}^{•-}$$
(8)$${{H_{2}\mathrm{PO}}_{4}}^{-}+SO_{4}^{•-}\to SO_{4}^{2-}+{{H_{2}\mathrm{PO}}_{4}}^{•-}$$
(9)$${{H_{2}\mathrm{PO}}_{4}}^{-}\to {{\mathrm{HPO}}_{4}}^{2-}+H^{+}$$
(10)$${{\mathrm{HPO}}_{4}}^{2-}+SO_{4}^{•-}\to SO_{4}^{2-}+{{\mathrm{HPO}}_{4}}^{•-}$$

### 3.4. Reusability Assessment of MoS_2_/MXene

In real application areas, the reusability of a catalyst is a crucial factor. The degradation rate of TC in the MoS_2_/MXene/PDS system served as an indication for repeated experiments using the catalysts. We conducted five tests with repeated cycles using the MoS_2_/MXene/PDS system. At the end of each cycle, MoS_2_/MXene was sonicated after being evenly distributed in a certain volume of ultrapure water. After natural sedimentation and centrifugation, the remaining concentration of TC in the supernatant was determined. Then, the preceding stages were repeated until there was no residual TC precipitated from the filtrate. Finally, the completely processed MoS_2_/MXene was dried overnight in a vacuum oven at 60 °C and ground to powder for the following cycle. As shown in Figure 6f, the removal rate of TC decreased steadily from 100% to 81.92% with an increase in cycle number. It can be inferred that MoS_2_/MXene is sufficiently reusable for the degradation of organic contaminants. Furthermore, the decreased performance might be due to a minor overflow of catalytic units contained in the catalyst [54]. The catalyst might also adsorb certain degradation intermediates, thus limiting the degradation effectiveness of TC [56]. Therefore, MoS_2_/MXene is practicable as an effective system for degrading TC by activating PDS, with some potential reusability.

### 3.5. Activation Mechanism of MoS_2_/MXene/PDS System

The distinct functions of reactive oxygen species generated in the MoS_2_/MXene/PDS system were investigated by employing free radical capture experiments. Some common scavengers and their rate constants are listed in Appendix A; ethanol (EtOH), tert-butyl alcohol (TBA), and p-benzoquinone (BQ) were chosen as the scavengers of SO_4_^•−^, •OH, and O_2_^•−^, respectively, for comparing their reaction constants [57]. As shown in Figure 7, different scavengers impacted the degradation efficiency of TC to varying degrees under the 1 mM addition condition. The inhibitory impact of EtOH in the MoS_2_/MXene/PDS system became apparent within the first 10 min of the reaction, demonstrating that SO_4_^•−^ produced by the activation of PDS via MoS_2_/MXene was primarily accountable for the degradation of TC in the primary phase of the reaction. The effect of TBA became increasingly apparent as the reaction continued, suggesting that when most of the PDS was engaged in the MoS_2_/MXene/PDS system, further degradation of TC depended on •OH generated by MoS_2_/MXene in the aqueous solution. The effectiveness of TC removal by EtOH and TBA decreased from 100% to 83.89% and 77.59%, respectively, after one hour of the reaction, demonstrating that SO_4_^•−^ and •OH were partially responsible for TC degradation. Additionally, the degradation efficiency of TC was somewhat reduced after BQ was introduced. After the reaction had proceeded for one hour, the degradation efficiency of TC reduced from 100% to 85.68%, indicating that O_2_^•−^ was extremely likely involved in the process. According to the results of the free radical capture experiments, it can be inferred that various reactive oxygen species play a significant role in the degradation of TC in the MoS_2_/MXene/PDS system, whose effect is in the order of •OH > SO_4_^•−^ > O_2_^•−^. Moreover, the non-significant inhibition further indicated the great stability and high efficiency of the MoS_2_/MXene/PDS system, as well as its potential for having outstanding adsorption capabilities in addition to acting as a catalyst for PDS activation.

To further explore the surface-active sites involved in the reaction, the elemental changes in both fresh and used MoS_2_/MXene, including valence and content, were characterized utilizing XPS analysis. As shown in Figure 7b, the peaks at 530.9, 458.7, 284.8, 228.4, and 161.9 eV corresponded to O 1s, Ti 2p, C 1s, Mo 3d, and S 2p, respectively. There was no significant change in the location or content of these peaks, indicating that MoS_2_/MXene possessed good stability (Appendix A). Moreover, the high-resolution XPS spectra of C 1s, O1s, Mo 3d, and S 2p are displayed in Figure 7c–f. As seen in Appendix A, the Mo^6+^ concentration increased from 0.07% to 0.86% after the reaction, whereas the Mo^4+^ content decreased from 5.93% to 4.54%, demonstrating that the slight oxidation of Mo^4+^ on the surface is a major factor for PDS activation. It is worth noting that the slight decrease in Mo^4+^ content might be attributed to the interconversion of MO^4+^ and Mo^6+^ [58]. Additionally, the Ti-O content increased from 5.61% to 8.57%, demonstrating that MXene and MoS_2_ were possibly engaged in the activation process of PDS and that their combined action could efficiently speed up this process. Meanwhile, the contents of C-O and C=O groups reduced from 16.52% and 5.93% to 9.88% and 3.99%, respectively. It was further revealed that both C-O and C=O groups were necessary for the activation of PDS and the generation of reactive oxygen species [59]. More interestingly, in the high-resolution XPS pattern of S, a novel signal peak at 164.0 eV was discovered, which was identified as bridging S_2_^2−^ at the apical of MoS_2_, suggesting a potential linkage reaction between pure MoS_2_ and further indicating the involvement of MoS_2_ in the oxidation reaction [60].

A suggested mechanism of PDS activation by MoS_2_/MXene for effective TC degradation was proposed after combining the above findings and analyses. Due to the combined piezoelectric and semiconducting properties of MoS_2_, effective removal of organic dyes could be accomplished in the dark (termed piezoelectric catalysis) [61]. A strong hydrogen peak was still detected in the solution after the removal of MoS_2_, which was likely due to the extended existence of a high concentration of •OH in the aqueous solution containing MoS_2_ [50]. This finding is also highly consistent with the results of the ESR detection. Therefore, the following equations (Equations (11)–(21)) were derived to account for the ROS generated during the oxidative degradation of TC, as well as the chemical reactions in the MoS_2_/MXene/PDS system, based on the experimental findings and previous reports [42,50,56,58,59]. Specifically, in the initial phase of oxidative degradation, electrons and holes are formed due to the piezoelectric catalysis and internal polarization of MoS_2_, which will stimulate the polarization decomposition of water molecules to produce •OH, and the above processes can be accelerated in acidic solutions exposed to air. Then, the high concentration of •OH promotes SO_4_^2−^ to generate SO4^•−^ in the aqueous solution, which rapidly attaches to organic molecules and triggers their oxidative breakdown [62]. Moreover, PDS can produce weak SO4^•−^ and O_2_^•−^ in an aqueous solution, while MoS_2_ can expedite this process and permit cyclic redox, considerably enhancing the utilization of the catalyst. Finally, TC is mineralized by a variety of ROS into different small molecules such as H_2_O and CO_2_.
(11)$$\mathrm{Mo}S_{2}(H_{2}O)\overset{\mathrm{Piezoelectric}}{\to }e^{-}+h^{+}$$
(12)$$H_{2}O+e^{-}\to •H+OH^{-}$$
(13)$$OH^{-}+h^{+}\to •\mathrm{OH}$$
(14)$$•\mathrm{OH}+SO_{4}^{2-}\to SO_{4}^{•-}+OH^{-}$$
(15)$$SO_{4}^{•-}+H_{2}O\to •\mathrm{OH}+SO_{4}^{2-}+H^{+}$$
(16)$$S_{2}O_{8}^{2-}+2H_{2}O\to HO_{2}^{-}+2SO_{4}^{2-}+3H^{+}$$
(17)$$S_{2}O_{8}^{2-}+HO_{2}^{-}\to SO_{4}^{•-}+SO_{4}^{2-}+O_{2}^{•-}+3H^{+}$$
(18)$$HO_{2}^{-}\to H^{+}+O_{2}^{•-}$$
(19)$${\mathrm{Mo}}^{4+}+{2S}_{2}O_{8}^{2-}\to {\mathrm{Mo}}^{6+}+2SO_{4}^{•-}+2SO_{4}^{2-}$$
(20)$${\mathrm{Mo}}^{6+}+{2S}_{2}O_{8}^{2-}+2H_{2}O\to {\mathrm{Mo}}^{4+}+2SO_{5}^{•}+2SO_{4}^{2-}+4H^{+}$$
(21)$$•\mathrm{OH}+SO_{4}^{•-}+O_{2}^{•-}+\mathrm{TC}\to \cdots \to CO_{2}+H_{2}O$$

### 3.6. Possible Degradation Pathways

To further clarify the degradation mechanism of TC, liquid chromatography–mass spectrometry (LC-MS) was used to identify the intermediates present in the MoS_2_/MXene/PDS system at different times. Appendix A and Appendix A show the m/z values and supposed structure of TC degradation intermediates at different times in the presence of the MoS_2_/MXene catalyst, and signal peaks with different m/z values can be observed for these intermediates. Based on the LC-MS data and previous reports, three possible degradation pathways of TC were proposed (Figure 8). In pathway I, the original TC molecule (*m*/*z* = 446) is converted to intermediate P11 (*m*/*z* = 460) via dehydration, which is followed by deamidation and dealkylation to yield intermediate P12 (*m*/*z* = 374). It is then converted to intermediates P13 *m*/*z* = 302) and P14 (*m*/*z* = 242) through oxidative ring-opening reaction and gradual decarbonylation. Finally, these intermediates are further oxidized to small molecules P15 (*m*/*z* = 240), P16 (*m*/*z* = 220), P17 (*m*/*z* = 228), and P18 (*m*/*z* = 99). In pathway II, the original TC molecule is first converted to intermediate P21 (*m*/*z* = 416) via demethylation. It then undergoes a dealkylation reaction to produce P22 (*m*/*z* = 376), and further deamidation and ring-opening reactions produce intermediate P23 (*m*/*z* = 338). Finally, it gradually forms small organic molecules via decarbonylation and oxidation reactions. Alternatively, P21 is slowly converted to other small molecules, namely, P24 (*m*/*z* = 360), P25 (*m*/*z* = 211), and P26 (*m*/*z* = 121), via progressive oxidation. In pathway III, the original TC molecule is first dehydrated and converted to intermediate P31 (*m*/*z* = 427). It can then undergo demethylation, dealkylation, and ring-opening reactions to produce P32 (*m*/*z* = 319). With further oxidation, other organics, namely, P33 (*m*/*z* = 279), P34 (*m*/*z* = 207), and P35 (*m*/*z* = 181), can be formed. Moreover, P31 can form intermediate P36 (*m*/*z* = 301) via a ring-opening reaction on the left side and hydroxylation and then intermediate P37 (*m*/*z* = 274) via deamidation. Finally, with further oxidative degradation, it can be transformed into intermediates P38 (*m*/*z* = 131) and P39 (*m*/*z* = 112). When MoS_2_/MXene is present in the system, TC can eventually be mineralized into inorganic substances, such as CO_2_ and H_2_O, with the oxidation of various ROS.

## 4. Conclusions

In this study, MoS_2_/MXene nanocomplexes were prepared in a simple one-step hydrothermal approach utilizing Ti_3_C_2_ MXene as a substrate, which was employed as an effective catalyst for the activation of PDS to effectively degrade TC. The characterization of MoS_2_/MXene was carried out using SEM, FTIR, and XPS, and it was observed that a stable 3D intercalation structure was formed through the self-assembly of MoS_2_ with MXene under solvent heat. This not only resulted in an expanded layer spacing and increased the number of surface-active sites for the catalyst, but it also successfully prevented MXene from being oxidized at high temperatures and inhibited MoS_2_ from clustering. More interestingly, MoS_2_/MXene performed better catalytically than individual MXene and MoS_2_, implying that tight contact between MoS_2_ and MXene occurred. The •OH produced was crucial in the activation of PDS and the oxidative destruction of TC when MoS_2_/MXene was present in the aqueous solutions, particularly in acidic solutions. In addition, TC could be completely removed in about 50 min under the conditions of a catalyst dosage of 0.4 g/L, a PDS concentration of 0.4 mM, and pH = 5.0 at room temperature. Despite the interference of various inorganic anions, TC was removed from the MoS_2_/MXene/PDS system relatively successfully. Through repeated testing, the recyclability of MoS_2_/MXene was assessed, and the catalyst was shown to have excellent reusability performance. Furthermore, the mechanism of action of the MoS_2_/MXene/PDS system on the TC degradation process was determined by extrapolating the formation principle of ROS during the reaction process and examining the intermediates and potential degradation pathways with LC-MS. Therefore, this study demonstrated that MoS_2_/MXene is a promising composite catalyst for the removal and degradation of persistent organic pollutants in water, and it also expanded the application of MoS_2_-based, MXene-based, and MoS_2_/MXene composites in catalytic oxidation.

## Figures and Tables

**Figure 1 nanomaterials-14-00786-f001:**
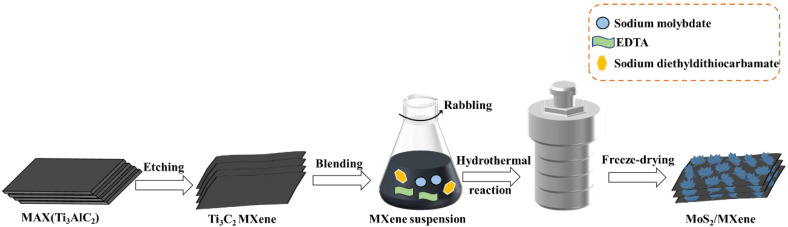
Schematic diagram of the synthesis of MoS_2_/MXene.

**Figure 2 nanomaterials-14-00786-f002:**
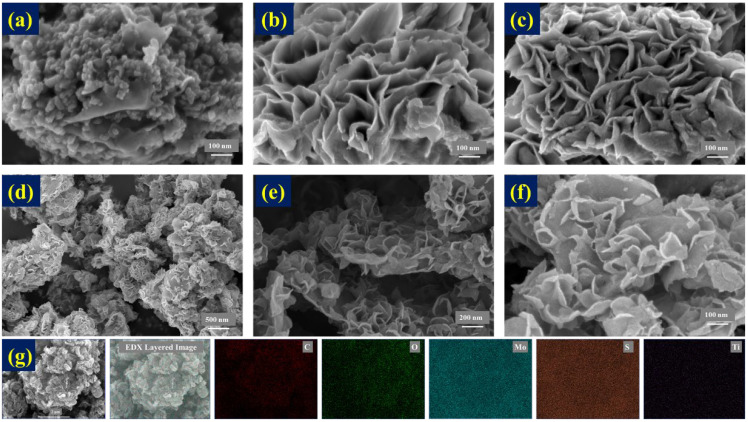
SEM images of (**a**) MXene, (**b**,**c**) MoS_2_, and (**d**–**f**) MoS_2_/MXene at different magnifications, and (**g**) EDX electron microscope scan images and elemental distribution.

**Figure 3 nanomaterials-14-00786-f003:**
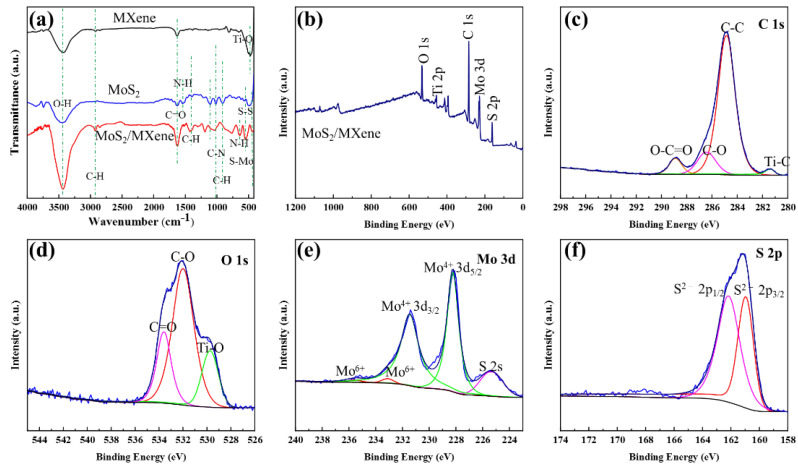
(**a**) FTIR spectra of MXene, MoS_2,_ and MoS_2_/MXene; (**b**) XPS survey spectra of MoS_2_/MXene, (**c**) C 1s, (**d**) O 1s, and (**e**) Mo 3d; and (**f**) S 2p spectra of MoS_2_/MXene.

**Figure 4 nanomaterials-14-00786-f004:**
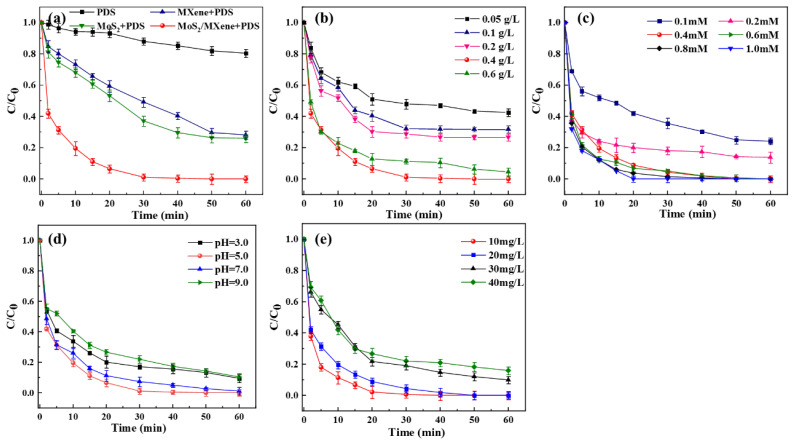
Effects of different factors on TC degradation: (**a**) material system; (**b**) catalyst dosage; (**c**) PDS concentration; (**d**) pH; and (**e**) TC initial concentration.

**Figure 5 nanomaterials-14-00786-f005:**
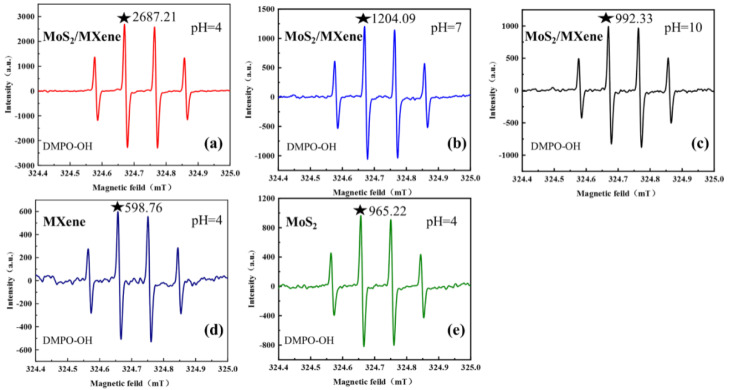
Intensity of DMPO-•OH signals in the presence of (**a**–**c**) MoS_2_/MXene, pH = 4, 7, or 9; (**d**) Ti_3_C_2_ MXene, pH = 4; and (**e**) MoS_2_, pH = 4.

**Figure 6 nanomaterials-14-00786-f006:**
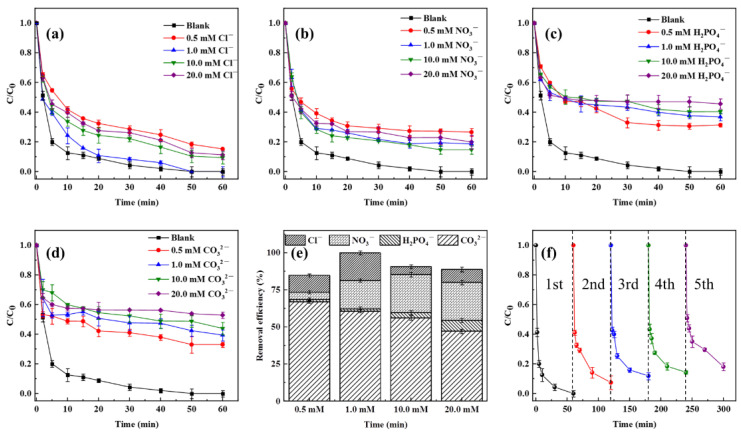
(**a**–**e**) Effects of different inorganic anions on the MoS_2_/MXene/PDS system, and (**f**) the degradation of TC in the recycling study of the MoS_2_/MXene/PDS system.

**Figure 7 nanomaterials-14-00786-f007:**
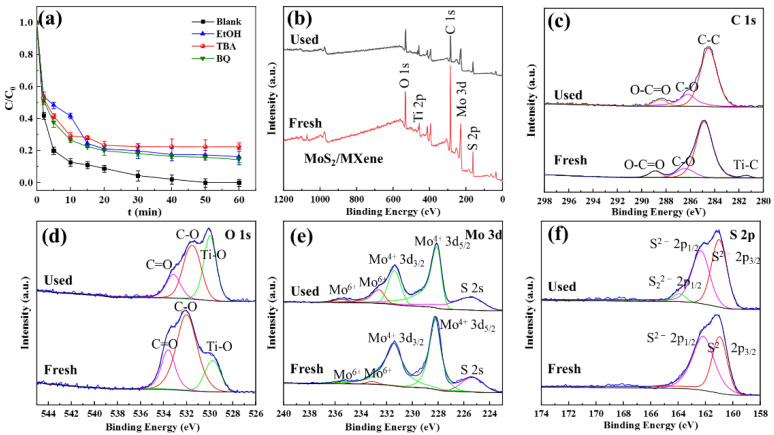
(**a**) Effects of different free radical bursting agents; (**b**) the XPS survey spectra of fresh and used MoS_2_/MXene, (**c**) C 1s, (**d**) O 1s, and (**e**) Mo 3d; and (**f**) S 2p spectra of fresh and used MoS_2_/MXene.

**Figure 8 nanomaterials-14-00786-f008:**
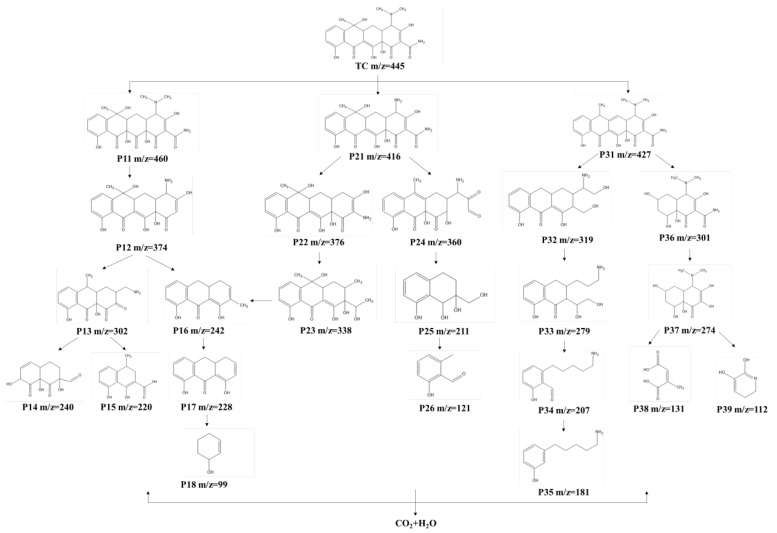
Possible degradation pathways of TC.

## Data Availability

Data are contained within the article and Supplementary Materials.

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
