# Peer review of "Enhancement of Peroxydisulfate Activation for Complete Degradation of Refractory Tetracycline by 3D Self-Supported MoS2/MXene Nanocomplex"

_nanomaterials, 2024, doi:10.3390/nano14090786_

Round 1

Reviewer 1 Report

Comments and Suggestions for Authors

Research article: Enhancement of peroxydisulfate activation for completely degradation of refractory tetracycline by 3D self-supported MoS2/MXene nanocomplex

In this article, the Authors the three-dimensional self-supported MoS2/MXene nanohybrid with expanded layer spacing synthesized by a facile one-step hydrothermal method and used to activate peroxydisulfate (PDS) for complete degradation of relatively stable antibiotic tetracycline. The research topic is novel and highly interesting; however, the execution is questionable, and the authors should answer some questions and make improvements to the manuscript prior to proposing to publish this article in the MDPI Nanomaterials.

A few remarks related to this work:

1)     The introduction part about MXenes is not sufficient. As the stability issue of MXenes mentioned, these papers should be mentioned as one of the easiest and most sensitive methods for Ti3C2Tx MXenes oxidation state evaluation and, in general, the internal structure analysis is Raman spectroscopy:

Chemosensors 2021, 9(8), 223; https://doi.org/10.3390/chemosensors9080223

https://doi.org/10.1021/acs.chemmater.0c00359

2)     The manuscript's MXenes formula Ti3C2 should be changed to Ti3C2Tx, where Tx – is the surface termination. Find more info: Nanomaterials 2024, 14(5), 447; https://doi.org/10.3390/nano14050447 ; ACS Nano 2019, 13, 8, 8491–8494 https://doi.org/10.1021/acsnano.9b06394

Moreover, it would be worth mentioning this information in the introduction part presenting MXenes.

3)     MM 2.5 section should be rewritten as it is unclear how the experiment's main part was done. How was PDS activated? Was it a photocatalytic process or how could the enhancement by adding MXenes-MoS2 be explained?

4)     As it is mentioned in introduction part: “Nevertheless, the hydrothermal/solvothermal techniques that are typically applied to produce MXene-based nanomaterials can accelerate MXene oxidation[27]. It is founded that Ti3C2 MXene is easily oxidized in the hydrothermal process due to the presence of dissolved oxygen, resulting in performance loss” how did this problem was solved in this manuscript?

5)     How the oxidation state of MXenes were evaluated?

6)     How the influence of MXenes in this study was evaluated? Authors should try to conduct one control experiments without MXenes and with only MoS2 synthesized by hydrothermal synthesis.

Comments on the Quality of English Language

Language should be double-checked for typos.

Author Response

Response to Reviewer 1 Comments

1. Summary

2. Questions for General Evaluation

Reviewer’s Evaluation

Response and Revisions

Does the introduction provide sufficient background and include all relevant references?

Must be improved

Are all the cited references relevant to the research?

Can be improved

Is the research design appropriate?

Can be improved

Are the methods adequately described?

Must be improved

Are the results clearly presented?

Can be improved

Are the conclusions supported by the results?

Can be improved

3. Point-by-point response to Comments and Suggestions for Authors

Comments 1: The introduction part about MXenes is not sufficient. As the stability issue of MXenes mentioned, these papers should be mentioned as one of the easiest and most sensitive methods for Ti3C2Tx MXenes oxidation state evaluation and, in general, the internal structure analysis is Raman spectroscopy: Chemosensors 2021, 9(8), 223; https://doi.org/10.3390/chemosensors9080223

https://doi.org/10.1021/acs.chemmater.0c00359

Response 1: Thanks for your valuable comments and thoughtful suggestions. We have add some discussion about the oxidation stability of MXenes in the introduction part. And the papers (https://doi.org/10.3390/chemosensors9080223; https://doi.org/10.1021/acs.chemmater.0c00359) as one of the easiest and most sensitive methods for Ti3C2TX MXenes oxidation state evaluation were also mentioned as references. The following sentence was added to the introduction. ‘However, the fabrication of devices and functional coatings based on Ti3C2TX remains challenging as they are susceptible to chemical degradation through oxidation to TiO2.’ (page 2, lines 86-89 in the revised manuscript).

Comments 2: The manuscript's MXenes formula Ti3C2 should be changed to Ti3C2Tx, where Tx – is the surface termination. Find more info: Nanomaterials 2024, 14(5), 447; https://doi.org/10.3390/nano14050447 ; ACS Nano 2019, 13, 8, 8491–8494 https://doi.org/10.1021/acsnano.9b06394. Moreover, it would be worth mentioning this information in the introduction part presenting MXenes.

Response 2:  Thanks to your professional opinion, we have changed the MXenes formula Ti3C2 to Ti3C2Tx in the whole manuscript and aslo revisited the introduction. The following sentence was added to the introduction. ‘It is mainly composed of transition metal carbides, and the chemical formula is commonly expressed as Mn+1XnTx, where M is a transition element (e.g., V, Mo, Ti, etc.), X is C or N, and Tx stands for surface terminations. Among the various MXene materials, Ti3C2Tx is the most widely used.’(page 2, lines 82-86 in the revised manuscript).

Comments 3: MM 2.5 section should be rewritten as it is unclear how the experiment's main part was done. How was PDS activated? Was it a photocatalytic process or how could the enhancement by adding MXenes-MoS2 be explained?

Response 3:  We appreciate your careful review of our paper. We have carefully revised this part. The following sentence will replace the original sentence.

‘All TC degradation experiments were carried out in 150 mL Erlenmeyer flasks at 25 °C under varying experimental conditions, which included the material system (MXene only, MoS2 only, and MoS2/Mxene), catalyst dosage, PDS consumption, pH, and initial concentration of TC.

Specifically, 50 mL of TC solution was placed in a 150 mL flask. After a certain amount of the catalyst material was added to the solution, which was stirred for 10 min, PDS of a certain concentration was added to initiate the degradation reaction. During the reaction process, 1.0 mL of the reaction solution was withdrawn at specific time intervals (10, 20, 30, 40, 50, and 60 min) with filtering. The recovered solution was filtered through a 0.22 μm filter and diluted to the standard curve concentration range with deionized water. The residual concentration was then detected at 356 nm using a UV–visible spectrophotometer (UV-2700, Shimadzu, Japan).

Additionally, 1) the impacts of inorganic anions (Cl, NO3, H2PO4, and CO32−), which are frequently present in natural water, were investigated in this work at a particular concentration of TC solution, a certain mass of catalyst, and a given molar mass of PDS solution. 2) Different concentrations of EtOH, TBA, and BQ were used as quenchers to study their effects on the oxidative degradation of TC.’

The •OH produced was crucial in the activation of PDS and the oxidative destruction of TC when MoS2/MXene was present in the aqueous solutions, particularly in acidic solutions.(page 4, lines 163-179 in the revised manuscript).

How was PDS activated, Was it a photocatalytic process or how could the enhancement by adding MXenes-MoS2 be explained?

Response: Combined with the results of free radical capture experiments and XPS survey spectra of fresh and used MoS2/MXene, we deduce that the redox of Mo4+, oxygen-containing functional groups such as Ti-O, -OH, C-O and C=O play an important role in the activation process of PDS as well as in the removal of TC.

As the results of ESR, ‘under the same circumstances, the intensity of DMPO-•OH signal was significantly lower than that of the MoS2/MXene nanocomposite when only MXene or MoS2 was present.’ we deduce that ‘the formation of particular heterojunctions between MoS2 and MXene enhanced the nanocomposite’s ability to transmit electrons, thus promoting the generation of •OH [41, 51], which potentially stimulated PDS activation and enhanced TC degradation in turn.’(page 8, lines 320-323 in the revised manuscript).

Comments 4: As it is mentioned in introduction part: “Nevertheless, the hydrothermal/solvothermal techniques that are typically applied to produce MXene-based nanomaterials can accelerate MXene oxidation[27]. It is founded that Ti3C2 MXene is easily oxidized in the hydrothermal process due to the presence of dissolved oxygen, resulting in performance loss” – how did this problem was solved in this manuscript?

Response 4:  In this study, we designed one-step hydrothermal process to make MoS2 nanoflowers strongly coupled with Ti3C2TX MXene. In this method, a new sulfur source, sodium diethyldithiocarbamate trihydrate (DDC), and a capping agent, EDTA, were used to substantially widen the interlayer spacing of the nanocomposite and to gain abundant surface-active sites, while the excellent chelating ability of DDC allowed the MoS2 precursor to be tightly coupled to Ti3C2TX MXene, thus effectively preventing the chemical reaction of Ti3C2TX MXene at the initial stage of the hydrothermal reaction. The characterization of MoS2/MXene was carried out using SEM, FTIR, and XPS, and it was observed that a stable 3D intercalation structure was formed through the self-assembly of MoS2 with MXene under solvent heat. This not only resulted in an expanded layer spacing and increased the number of surface-active sites for the catalyst, but it also successfully prevented MXene from being oxidized at high temperatures and inhibited MoS2 from clustering. In addition, the results of TC degradation experiment indicated that MoS2/MXene performed better catalytically than individual MXene and MoS2, implying that a tight contact between MoS2 and MXene occurred. The •OH produced was crucial in the activation of PDS and the oxidative destruction of TC when MoS2/MXene was present in the aqueous solutions, particularly in acidic solutions.

Comment 5: How the oxidation state of MXenes were evaluated?

Response 5:  Ti3C2Tx MXenes are known to undergo oxidation over time. Since MXenes derived from ceramics (MAX phase) through etching, it is the inevitable exposure of metal atoms on their surface and embedding of anions and cations. Because the as-obtained MXenes are always in a thermodynamically metastable state, they tend to react with trace oxygen or oxygen-containing groups to form metal oxides or degrade, leading to sharply declined activity and impaired performance.

The oxidation state of MXenes could be evaluated by Raman characterisation. As mentioned in these papers (https://doi.org/10.1021/acs.chemmater.0c00359; https://doi.org/10.3390/chemosensors9080223), Raman spectroscopy is a tool for detecting early signs of oxidation and sample degradation that are difficult to detect by other methods.

The oxidation state of MXenes could be evaluated by X-ray photoelectron spectroscopy (XPS) through elemental analysis (e.g. observing the high-resolution Ti2p XPS spectra, the oxidation of MXene to TiO2 can be determined by comparing the increase in the atomic percentage of Ti-O 2p3/2 in all fractions. In addition, the oxidation of MXenes can be evaluated by XRD and SEM. For example, the (002) peak of pure MXene in the XRD pattern disappears completely and the signal of TiO2 can be detected. Combined with the changes in the micromorphology of MXenes in SEM can also indicate the oxidation of MXenes.

In our present work, we designed one-step hydrothermal process to make MoS2 nanoflowers strongly coupled with Ti3C2TX MXene. This not only resulted in an expanded layer spacing and increased the number of surface-active sites for the catalyst, but it also successfully prevented MXene from being oxidized at high temperatures and inhibited MoS2 from clustering. The synthesised composites were characterised and analysed using SEM, EDX, FTIR and XPS, which could provide some information on the controlled oxidation of Ti3C2Tx MXenes. (e.g. through SEM, it was observed that numerous MoS2 nanoflowers were uniformly growing on the MXene surface. With further magnification, it could be clearly observed that an intercalation structure between MoS2 and MXene was formed, and the disappearance of TiO2 particles on the MXene surface suggested that the addition of MoS2 partially shielded the oxidation of MXene.)

Comment 6: How the influence of MXenes in this study was evaluated? Authors should try to conduct one control experiments without MXenes and with only MoS2 synthesized by hydrothermal synthesis?

Response 6:  Thanks for pointing that out. We are actually conduct the control experiments with MXene only, MoS2 only, and self-supported MoS2/MXene nanohybrid. We have mentioned in the MM 2.5 in the latest version. And in the Results and discussion part ‘3.2. Performance regarding TC degradation’ We have discussed the effects of different materials and systems (MXene only, MoS2 only, MoS2/MXene and nothing catalyst just PDS),and the results was shown in Fig.4a.

The following sentence has discussed the influence of MXenes on TC degradation experiment in this study. And of course, the influence of MXenes on the whole three-dimensional self-supported MoS2/MXene nanohybrid for the enhancement of peroxydisulfate activation were also evaluated through SEM, FTIR, and XPS analysis.

‘As shown in Fig. 4a, the effects of different materials or systems on TC degradation were studied under the same catalyst dosage or PDS concentration conditions. In the presence of PDS alone for 1 h, the degradation efficiency of TC was poor (18.07%), while the addition of MXene or MoS2 accelerated the degradation efficiency of TC (71.82% and 74.06%, respectively). More interestingly, the inclusion of MoS2/MXene significantly accelerated the degradation of TC, which could achieve 100% efficiency at around 50 minutes. In addition, it can be inferred from Fig. S2 that all four systems conformed to the pseudo-first-order kinetic equations when degrading TC, and the parameters are listed in Table S2. These results imply that the vertical development of MoS2 on MXene exposes more active sites, thus enhancing electron mobility across the composites and promoting the rapid degradation of TC.’(page 7, lines 253-263 in the revised manuscript).

In addition, Fig.4a also showed that the PDS activated by photocatalytic (visible photoactivily in our work, we did not manually add other light sources such as UV) generated less ROS than other materials/systems for degrading TC in our present work.

4. Response to Comments on the Quality of English Language

Point 1: Language should be double-checked for typos.

Response 1:    Thanks for your advice. We have submitted the manuscript to MDPI's professional editing team for English proofreading. All changes are highlighted in the latest version. Here is the English-Editing-Certificate.

5. Additional clarifications

The suggestions are very helpful for us to make improvements, and we have incorporated them into the revised paper. We have carefully studied the reviewers’ comments and suggestions and tried our best to revise our manuscript according to the comments.

Reviewer 2 Report

Comments and Suggestions for Authors

In the manuscript entitled "Enhancement of peroxydisulfate activation for completely degradation of refractory tetracycline by 3D self-supported MoS2/MXene nanocomplex", the use of MoS2/Mxene for the improvement of the tetracycline (TC) removal based on peroxydisulfate (PDS) has been discussed. 

The introduction is very clear and accurately describes the main issue, the target and the reasons behind the investigations performed and described in the manuscript. Synthesis of MoS2/MXene is accurately described and its characterization (SEM images clearly point to a stable intercalation structure; FTIR spectra also are very representative of the situation; chemical state was also assessed by using XPS) too (Figure 1 is nicely made). 

ESR examination helps to study the increase in OH radical generation under several pH conditions.

Factors (pH, catalyst concentration, PDS concentration and initial TC concentration, inorganic anions) affecting the tetracycline degradation are accurately exhibited in Figure 4 and Figure 6 and the outcomes are coherent with the thesis proposed in the paper.

Possible degradation pathways of tetracycline are also examined in the Figure 8 and the conclusion are well written. 

Nothing to say, this manuscript can be accepted in the present form

Author Response

Response to Reviewer 2 Comments

1. Summary

2. Questions for General Evaluation

Reviewer’s Evaluation

Response and Revisions

Does the introduction provide sufficient background and include all relevant references?

Yes

Are all the cited references relevant to the research?

Yes

Is the research design appropriate?

Ye

Are the methods adequately described?

Yes

Are the results clearly presented?

Yes

Are the conclusions supported by the results?

Yes

3. Point-by-point response to Comments and Suggestions for Authors

Comments 1: In the manuscript entitled "Enhancement of peroxydisulfate activation for completely degradation of refractory tetracycline by 3D self-supported MoS2/MXene nanocomplex", the use of MoS2/Mxene for the improvement of the tetracycline (TC) removal based on peroxydisulfate (PDS) has been discussed.

The introduction is very clear and accurately describes the main issue, the target and the reasons behind the investigations performed and described in the manuscript. Synthesis of MoS2/MXene is accurately described and its characterization (SEM images clearly point to a stable intercalation structure; FTIR spectra also are very representative of the situation; chemical state was also assessed by using XPS) too (Figure 1 is nicely made).

ESR examination helps to study the increase in OH radical generation under several pH conditions.

Factors (pH, catalyst concentration, PDS concentration and initial TC concentration, inorganic anions) affecting the tetracycline degradation are accurately exhibited in Figure 4 and Figure 6 and the outcomes are coherent with the thesis proposed in the paper.

Possible degradation pathways of tetracycline are also examined in the Figure 8 and the conclusion are well written.

Nothing to say, this manuscript can be accepted in the present form

Response 1:  Thank you for your compliments. We appreciate your recognition of our work. In addition, to improve the readability of the article, we have submitted the manuscript to MDPI's professional editing team for English proofreading. All changes are highlighted in the latest version. Here is the English-Editing-Certificate.

4. Response to Comments on the Quality of English Language

Point 1: I am not qualified to assess the quality of English in this paper

Response 1:    We have submitted the manuscript to MDPI's professional editing team for English proofreading. All changes are highlighted in the latest version.

5. Additional clarifications

The suggestions are very helpful for us to make improvements, and we have incorporated them into the revised paper. We have carefully studied the reviewers’ comments and suggestions and tried our best to revise our manuscript according to the comments.

Round 2

Reviewer 1 Report

Comments and Suggestions for Authors

Dear Authors,

thank you for the improvements to the manuscript. The manuscript can be proposed to the Editors for publication in Nanomaterials.